# Comparison of Paliperidone Palmitate from Different Crystallization Processes and Effect on Formulations In Vitro and In Vivo

**DOI:** 10.3390/pharmaceutics14051094

**Published:** 2022-05-20

**Authors:** Junfeng Shi, Dan Wang, Yang Tian, Zengming Wang, Jing Gao, Nan Liu, Xiang Gao, Aiping Zheng, Hui Zhang, Meixian Xiang

**Affiliations:** 1School of Pharmaceutical Sciences, South-Central University for Nationalities, Wuhan 430074, China; 18229969681@163.com; 2Institute of Pharmacology and Toxicology, Academy of Military Medical Sciences, Academy of Military Sciences, Beijing 100850, China; wanfdan5775@163.com (D.W.); tianyang1127@126.com (Y.T.); wangzm.1986@163.com (Z.W.); gjsmmu@126.com (J.G.); wowlinan@sohu.com (N.L.); gaoxiang609@163.com (X.G.); apzheng@163.com (A.Z.)

**Keywords:** paliperidone palmitate, crystallization processes, surface free energy, crystallinity, stability, pharmacokinetics

## Abstract

The quality of active pharmaceutical ingredients (APIs) is an important factor which can affect the safety and efficacy of pharmaceuticals. This study was designed to investigate the nature of paliperidone palmitate (PP) obtained by different crystallization processes, then compare the characteristics between test formulations which prepared PP of different crystallization and reference formulations (Invega Sustenna^®^) in vitro and in vivo. Two different PPs, namely PP-1 and PP-2, were prepared by different crystallization methods. Contact angle, morphology, and crystallinity of the PPs were characterized. Taking the particle sizes and distribution of Invega Sustenna^®^ as reference, test formulations were prepared by the wet milling method using either a PP-1 or PP-2 sample. Their release behavior, stability in vitro, and pharmacokinetics in vivo were subsequently investigated. The results indicated that PP-2 had a higher surface free energy (SFE). More small particles were attached to the PP-1 surface under the influence of crystallization temperature. Different crystallization processes did not change the crystal of PP, but changed the crystallinity of PP. There was no obvious difference in in vitro releases between test formulations. However, the stability and state of formulation containing PP-2 were better compared to formulations containing PP-1, indicated by differences in crystallinity and SFE. Meanwhile, pharmacokinetic in vivo results demonstrated that the pharmacokinetic profiles and parameters of formulation containing PP-2 and Invega Sustenna^®^ tended to be consistent, but those of formulations containing PP-1 were significantly different from those of formulations containing PP-2 or Invega Sustenna^®^, and there was burst release phenomenon of formulations containing PP-1 in rats. PP made by different crystallization processes could induce changes in appearance, SFE, and crystallinity, and further affect the stability, state, and pharmacokinetic in vivo formulation.

## 1. Introduction

Paliperidone palmitate (PP, Figure 1) is a second-generation antipsychotic agent that has been used effectively in the treatment of schizophrenia [1]. PP, a prodrug of paliperidone, hydrolyzes into palmitic acid and paliperidone (9-hydroxyrisperidone) after administration. Paliperidone is pharmacologically active substance, and it achieves its role by blocking dopamine 2 receptors, which in turn reduces symptoms of psychosis and has a stabilizing effect on the affective symptoms. Furthermore, it enhances dopamine release in certain brain regions and in turn reduces motor side effects and possibly improves cognitive and emotional symptoms by blockage of serotonin 2A receptors. Antagonism at the serotonin 7 receptor is proposed as a possible contributor to the antidepressant actions [2,3]. Antipsychotic medication plays a prominent role in schizophrenia symptom management.

Long-acting injections (LAI) are parenteral agents, which are designed to deliver the active pharmaceutical ingredient (API) at a controlled rate to achieve prolonged therapeutic exposure [4]. At present, four long-acting anti-schizophrenia products developed by the Janssen Pharmaceutical Company have been approved by the Food and Drug Administration (FDA). Due to the advantages of low dosage and low recurrence rate, it is favored by most patients. Investigations have indicated that the global sales of paliperidone palmitate injection have been increasing steadily year by year since its launch. According to the instructions and data [5], prior to Invega Trinza^®^ and Invega Hafyera^®^, patients must receive Invega Sustenna^®^ for a period. Therefore, Invega Sustenna^®^ plays a pivotal role in the treatment of schizophrenia. The three long-acting anti-schizophrenia products of paliperidone palmitate are injectable aqueous suspension formulations [4]. It is a well-known fact that injectable formulations bare some advantages over oral administration, such as the avoidance of first-pass metabolism, the certainty of delivery of the therapeutic agent, and the enhanced patient adherence [6]. More importantly, the aqueous suspension formulations have the advantages of less excipience, fewer problems of encapsulation rate, fewer issues related to drug loading. Therefore, the API plays an important role in injectable aqueous suspension formulations compared with ordinary formulations. The physical and chemical properties or other quality characteristics of the API will directly determine the effectiveness, safety, and other key properties of the final product to a large extent. Particularly for insoluble drugs, the crystallinity, particle size distribution, and other factors of the API can directly affect the production and quality of the final product. Therefore, adequate comparison, analysis, research, and evaluation of the quality characteristics of APIs from different sources should be performed prior to drug development.

PP is processed by a solvent crystallization method, so the crystallization process is particularly critical. Previous studies have indicated that the crystallization process may affect the crystallinity and particle microstructure of drugs [7,8]. The nature of paliperidone palmitate requires in-depth characterization. No single characterization technique can give the full picture, so a suite of complementary approaches is often required. Contact angle measurement is one of the most used techniques for solid surface characterization, and the surface free energy (SFE) of solids can be calculated using contact angle values [9]. The contact angle is formed by the liquid in contact with the surface of solids, helping to understand the surface properties of the solids [10,11]. Scanning electron microscopy (SEM) is widely used to observe the external morphology of substances. X-ray powder diffraction (XRPD) is an indispensable tool for the characterization of solid dispersions [12]. Previous studies have reported that the XRPD technique was used to confirm the presence of amorphous drugs in solid dispersions [13,14,15]. Differential scanning calorimetry (DSC) is one of the most important thermal analysis methods, which has been used for routine analysis in the pharmaceutical industry [16]. A molecular-level study in the solid dispersion such as glass transition, crystallization, polymorphic transition, and molecular mobility can be acquired using DSC [17]. Infrared spectrum (IR) also has been used for a range of pharmaceutical applicants which can provide various information on molecular-level attributes such as surface properties, degree of amorphization, phase transitions, and crystallization [18]. In addition, water vapor sorption [19], Solid-State nuclear magnetic resonance (ssNMR) [20], and inverse gas chromatography (IGC) [21] have been frequently used to study the behaviors of amorphous and crystalline materials.

In the present study, contact angle, SEM, DSC, XRPD, and IR were used to analyze the macroscopic and microscopic characteristics of PP obtained by different crystallization methods, and the test formulations’ in vitro and in vivo behaviors were further evaluated. Theoretically, test formulation and reference formulation should be pharmaceutically and therapeutically equivalent, and the former can play a role of alternative treatment to a certain extent [22]. Moreover, evaluations of test formulations in vitro and in vivo were compared with the reference preparation, which can distinguish whether test formulations and reference formulations can be pharmaceutically equivalent. As the active substance of the formulation, the physicochemical properties of PP likely play a crucial role in vitro and in vivo, so it is necessary to have a detailed understanding of its characteristics.

## 2. Materials and Methods

### 2.1. Materials

Invega Sustenna^®^ (reference formulation, KFB5Y00) was obtained from Janssen Pharmaceutica N.V. Tween 20 and Poly(ethylenglycol) 4000 (PEG4000) were provided by Nanjing Well Pharmaceutical Co., Ltd. (Nanjing, China). Sodium dihydrogen phosphate and dibasic sodium phosphate were supplied by Chengdu Huayi Pharmaceutical Accessories Manufacturing Co., Ltd. (Chengdu, China). Citric acid was purchased from Sinopharm Chemical Reagent Co., Ltd. (Shanghai, China). Sodium hydroxide was obtained from Sichuan Jinshan Pharmaceutical Co., Ltd. (Meishan, China). Diazepam was purchased from Shandong Xinyi Pharmaceutical Co., Ltd. (Dezhou, China). Methanol, acetonitrile, N, N-dimethylformamide, and tetrahydrofuran of chromatographic grade were purchased from Fisher Scientific (Pittsburgh, PA, USA).

### 2.2. Crystallization Processes of Paliperidone Palmitate

The crude product of PP was synthesized by palmitic acid and paliperidone, and the last step of PP preparation was consecutive solvent crystallization to obtain a pure drug.

PP-1 was obtained by two consecutive crystallizations of ethyl acetate and ethanol, respectively. The 100 g crude product of PP was taken, and one liter of ethyl acetate was added with constant stirring for crystallization. Dissolution of all solid particles was achieved by heating to 80 °C. The crystallizer was cooled to 20 °C and held for 30 min, establishing supersaturation. The cake became a slurry in 10 times the volume of ethanol (*w*/*v*). We heated the solution to 80 °C to dissolve all solid particles. The crystallizer was cooled to 20 °C at room temperature, holding for 30 min. The solid product was isolated by vacuum filtration. PP-2 was obtained by three successive crystallizations of ethyl acetate, ethyl acetate, and ethanol, respectively. The twice-crystallization process of ethyl acetate was the same as above. The difference was the cooling step of ethanol crystallization, and the cooling step was not at a temperature but at a rate of 3 °C/h.

### 2.3. Characterization of Paliperidone Palmitate

#### 2.3.1. Contact Angle Evaluation

In this study, Tween 20 was used as a surfactant to enhance wetting ability of PP and reduce its surface tension. Tween 20 solution (0.3%, *w/v*) was prepared and its surface tension was evaluated by the suspension drop method, which was conducted by a Droplet Shape Detector (Kruss, DSA25, Stuttgart, Germany) at room temperature.

PP (5 g) was compressed to form a disk. Then, the contact angle between 0.3% Tween 20 solution and PP was measured by a drop method, which also conducted by the Droplet Shape Detector at room temperature.

#### 2.3.2. Granular Evaluation

The morphologies of PP-1 and PP-2 were observed by SEM (Akishima, JEOL, Osaka, Japan) with energy dispersive spectroscopy (EDS). The samples were measured with the accelerating voltage of 3.0 kV.

#### 2.3.3. Crystallinity Evaluation

##### Differential Scanning Calorimetry

The thermal characteristics of PP-1 and PP-2 were characterized by differential scanning calorimetry (DSC) (TA, Q2000, New Castle, DE, USA). The samples were sealed in an aluminum pan and the heating temperature changed from 60 to 140 °C with a heating rate of 10 °C/min in the atmosphere of nitrogen.

##### Infrared Spectroscopy

The chemical structures of PP-1 and PP-2 were investigated by IR (Scientific, Nicolet is5, Waltham, MA, USA). The samples were made into tablets with KBr and the scan range was from 400 cm^−1^ to 4000 cm^−1^ with a resolution of 4 cm^−1^.

##### The Crystalline Properties

The crystalline properties of PP-1 and PP-2 were analyzed by XRPD (Bruker, D2, Karlsruhe, Germany) coupled with a KFL CU 2K source of radiation. The sample was scanned at 2*θ* range from 4° to 50° with a step angle of 0.02° and counting time of 1 s/step, and the acquisition time of each spectrum was 39 min.

### 2.4. Preparation of Test Formulations

The test formulations were prepared by wet milling, which was performed in an agitator bead mill (DYNO^®^-MILL MULTI LAB, WAB, Basel, Switzerland). PP was characterized as practically insoluble in water (intrinsic solubility below 0.1 µg/mL) [23], so it was prepared into suspension by a wetting agent. First, Tween 20 (3%, *w/v*) was dissolved in water. After complete dissolution, PP was dispersed in Tween 20 solution to obtain a PP-concentrated suspension (500 g). The concentrated suspension was homogenized by the homogenizer (FLUKO, FM200A, Shanghai, China) for 10 min to obtain a uniformly concentrated suspension. Next, the concentrated suspension was pumped into the chamber to mill in passage mode. Finally, the PP-concentrated suspension after milling was diluted by saline solution, which was composed of PEG4000, sodium dihydrogen phosphate, dibasic sodium phosphate, citric acid, and sodium hydroxide [24] to obtain the formulation containing PP. The preparation progress of PP formulation is proposed in Figure 2. In this study, formulation was prepared using either a PP-1 or PP-2 sample.

### 2.5. Characterization of Formulations

#### 2.5.1. Particle Size Analyses

The particle sizes of PP formulations were measured by laser diffraction (LD) using a Malvern Mastersizer 2000 Instrument (Malvern, Worcestershire, UK). First, the formulation was diluted 100 times with water. The determination parameter setting of water was selected as the dispersion medium and the pump speed was set to 1250 rpm. The diluted formulation was added drop by drop to the water until the obscuration reached 6.8~7.2%. Each sample was measured in triplicate.

#### 2.5.2. Dissolution Evaluation

##### In Vitro Release

The dissolution was performed for formulation containing PP-1 or PP-2 with the paddle method (Tianda Tianfa Technology Co., Ltd., RC806D, Tianjin, China). The temperature was 25 °C and the paddle speed was 50 rpm. According to the FDA dissolution method library, the dissolution medium was 900 mL of 0.001 mol/L hydrochloric acid solution containing 0.489% Tween 20. 0.5 mL of suspension was injected directly into the dissolution medium, and 4 mL of the sample was withdrawn at 1.5, 5, 10, 15, 20, 30, 45, 60, and 90 min. All samples were filtered using 0.2 μm nylon membranes. In addition, the dissolution of Invega Sustenna^®^ was determined as the above.

The PP content in the samples was measured by HPLC (Thermo, UltiMate3000, Waltham, MA, USA) with an ultraviolet detector at the detection wavelength of 280 nm. Isocratic elution was conducted for chromatographic separation with methanol-acetonitrile (20:80, *v/v*) as the mobile phase A and 0.01 mol/L ammonium acetate as the mobile phase B. The flow rate was 1.0 mL/min. The column and auto-sampler temperatures were 35 °C and 10 °C, respectively, and the sample volume injected was 20 μL.

##### Similarity Factor (*f*_2_)

Similarity measurement is a requirement for many pharmaceutical industries to develop new drug applications [25], and Moore and Flanner proposed the similarity factor (*f*_2_) to compare pairwise dissolution profiles [26]. Therefore, the similarity factor method was used to assess the dissolution rate profiles in this study, which provided percentage dissolution between the two curves to measure similarity between two formulation release profiles [27]. The Equation (1) of similarity factor *f*_2_ is as follows:(1)$$f_{2\;}=50log\left\{{\left(1+\frac{1}{n}\overset{n}{\underset{j=1}{\sum }}{\left(R_{j}-T_{j}\right)}^{2}\right)}^{-0.5}\times 100\right\}\;\;\;\;\;\;\;\;\;\;\;\;\;$$
where *n* is the number of dissolution points, *R_j_* is the dissolution value of reference formulations at time t, and *T_j_* is the dissolution value of test formulations at time t.

### 2.6. Stability of Test Formulations

Test formulations were sealed and stored at room temperature for a period and particle size status was observed regularly. The measurement of particle size was the same as Section 2.5.1. Simultaneously, the re-dispersibility was evaluated by re-shaking to observe the state of test formulations.

### 2.7. Pharmacokinetics In Vivo

#### 2.7.1. Animal

Male Sprague Dawley (SD) rats weighing 200~210 g were purchased from Weitong Lihua Experimental Animal Technology Co., Ltd., Beijing, China (experimental animal license No. SCXK-(Beijing) 2016-0011). Rats were raised at room temperature (25 ± 1 °C) and relative humidity 40~70% for 3 days and were provided free access to food and water. All studies followed the Guide for the Care and Use of Laboratory Animals.

#### 2.7.2. Pharmacokinetics Study

The 15 SD rats were randomly divided into three groups. The first group was given the formulation containing PP-1, the second group was injected with the formulation containing PP-2, and the last group was given Invega Sustenna^®^. The rats were fasted overnight prior to experiments and were allowed free access to water. The rats were weighed before administration of PP formulations. The rats were given intramuscular (i.m.) of 193.3 mg/kg formulation containing PP-1, formulation containing PP-2, and Invega Sustenna^®^, respectively. Approximately 0.2 mL of orbital blood was collected at 1, 2, 3, 6, 8, 24, 48, 72, 120, 144, 216, 288, 384, 504, 672, and 840 h after administration. Then, blood was centrifuged at 5000 rpm for 5 min, and plasma was separated and stored at −20 °C until the analysis.

#### 2.7.3. Measurement of Paliperidone in Blood

A 40 μL plasma sample was taken, and 10 μL internal standard (diazepam) solution (500 ng/mL) and 10 μL methanol were added before the mixture was vortexed for 1 min. Thereafter, 500 μL diethyl ether was added to extract paliperidone. The mixture was vortexed for 3 min and then centrifuged at 13,000 rpm for 10 min. The supernatant was transferred into another sampling tube and evaporated to dry. The residue was dissolved with 100 μL methanol-water (80:20, *v*/*v*). The mixture was then centrifuged at 13,000 rpm for 10 min. The supernatant was used to be determined, and the detection volume was 5 μL [28].

The samples were assayed using a liquid chromatography–mass spectrometer (LC/MS) (Agilent 6460, Palo Alto, CA, USA), which was equipped with an ionization source and used in the positive ion mode. A Kinetex^®^ C_18_ column (50 × 2.1 mm, 2.6 μm; Agela Technologies, Tianjin, China) was employed to analyze the samples. The mobile phase A was acetonitrile and the mobile phase B was water with 0.1% ammonia, and isocratic elution was conducted for chromatographic separation. The column temperature was 35 °C. In addition, the collision energies of paliperidone and diazepam were 30 V and 33 V, respectively. For the quantification, the multiple-reaction monitoring (MRM) transition pairs of *m*/*z* 427.1→207.0 and *m*/*z* 285.1→193.1 were employed for paliperidone and diazepam, respectively.

### 2.8. Statistical Analysis

Omnic 8.2 was used for baseline correction and peak calibration of the IR original data, and peak fit was used for fitting analysis. Igor Pro software was used to perform peak fitting for XRPD data, and the Gaussian peak shape function was adopted. The pharmacokinetics data were processed using the Phoenix pharmacokinetic software and a non-compartment model, and all the results were presented as the mean ± SD (standard deviation).

## 3. Results and Discussion

### 3.1. Characterization of Paliperidone Palmitate

#### 3.1.1. Surface Free Energy Evaluation

The surface tension of Tween 20 solution is exhibited in Figure 3. The surface tension of Tween 20 was 39.8 mN/m, which was significantly less than the surface tension of water, 72.8 mN/m. This indicated that Tween 20 can decrease the interfacial tension of water.

Wetting performance is one of the most basic properties of surfactants [29]. A small amount of surfactant is added to the liquid. The surfactant forms a directional adsorption layer on the surface of the low-energy surface solid, hydrophobic toward the solid surface and hydrophilic outward, changing the nature of the solid surface. Conversely, the surfactant reduces the surface tension of the liquid and enhances the wetting ability [30].

The contact angles of Tween 20 on PP-1 and PP-2 are displayed in Figure 4; the results were 64.5° and 55.6°, respectively. The contact angle reflects the wetting of a liquid on a solid surface. The wetting equation proposed by T. Young revealed the specific relationship between the contact angle and the surface free energy, which indicates that the higher the solid surface energy is, the smaller the contact angle is and the more easily the wetting occurs [31]. This indicated that the surface free energy of PP-2 was higher, and it is easier wetted by Tween 20 solution. It was observed that PP-2 was more easily wetted by Tween 20 solution than PP-1 and had a better dispersion in Tween 20 solution during preparation. Interestingly, the measurement results of contact angles were consistent with the experimental phenomenon in the preparation process.

#### 3.1.2. Granular Evaluation

The morphologies of PP-1 and PP-2 are displayed in Figure 5. The images indicate an irregular platter shape for PP-1 and PP-2. As can be observed from the images, PP-2 has a smoother surface than PP-1. PP-1 has more smaller particles attached to its surface.

In the process of drug crystallization, the cooling speed plays a significant role in the quality of drug. The formation and structure of the drug crystal would indicate different states with the change of cooling rate. Meanwhile, the supersaturation of the solution changes with different cooling rates. The supersaturation of solution is related to the formation rate of crystal nuclei and the growth rate of crystal. The larger the supersaturation is, the more crystal nuclei are generated and the smaller the crystal size is. In the crystallization process of PP-1, the saturation increased due to the higher cooling rate, and more small particles were produced. Therefore, PP-1 and PP-2 had obviously different appearances.

#### 3.1.3. Crystallinity Evaluation

##### DSC

The DSC thermograms and melting parameters of PP-1 and PP-2 are portrayed in Figure 6. PP-1 and PP-2 indicated endothermic peak maximums at 117.2 °C and 118.6 °C, respectively. Furthermore, the proportionalities of melting heat to the amounts of PP for PP-1 and PP-2 were 82.6 and 80.4 mJ/mg, respectively. It is well known that the melting point of organic materials decreases with the decrease in particle size [32,33]. SEM results indicated that PP-1 had a smaller particle size than that of PP-2, so the melting point of PP-1 was lower than that of PP-2. In addition, the higher melting point caused higher crystallinity. However, further study will be conducted via IR and XRPD.

##### IR

The IR spectrum of PP-1 and PP-2 are exhibited in Figure 7. IR could indicate the differences in the infrared characteristic absorption spectra among functional groups in a material structure [34]. The IR characteristic peaks of paliperidone palmitate were relatively complex, with a total of 17 obvious characteristic absorption peaks, mainly including hydroxyl, methyl, methylene, and nitrogen-containing functional groups.

Paliperidone palmitate has an obvious characteristic absorption peak above 3000 cm^−1^ which is wide in shape indicating that there might be some water in the samples, thus the stretching vibration of hydroxyl group appeared [35]. Characteristic peaks of No. 16~13 are antisymmetric and symmetric stretching vibrations of methyl and methylene [35,36]. The characteristic peaks of No. 11 and 12 are stretching vibration peaks of C=O, which can be divided into two obvious peak shapes. The larger-wavelength characteristic peak is from the carbonyl structure of palmitic acid structure, while the smaller wavelength characteristic peak may be caused by stretching vibrations of C=O on the ring. The characteristic peaks of No. 9 and 10 are typical C=C characteristic absorption peaks of aromatic compounds. The characteristic peak of No. 8 is related to the shear vibration of CH_2_ [37]. The characteristic peak of No. 7 usually appears in hydrocarbon ester compounds, mainly consisting of ring structure and C-N stretching vibration peaks in acid ester groups [38,39]. The characteristic peak of No. 6 is related to the vibration of CH_3_ [36]. The characteristic peak of No. 5 is the stretching vibration peak of C-O-C, a typical ether functional group [35]. The band of 700 cm^−1^~1060 cm^−1^ is mainly the characteristic region of aliphatic C-skeleton, so peak 1 and peak 2 may be related to C-C/C-H structures, which are derived from palmitic acid and benzoxazole, respectively [37]. The characteristic peak of No. 3 is a typical characteristic absorption peak of C-F bonds. The characteristic peak of No. 4 is mainly located in the characteristic region of oxygen-containing functional groups. In previous studies, the stretching vibration of the ether bond mainly occurred in this region [35].

The fitting diagrams of each functional group in PP-1 and PP-2 are displayed in Figure 8. Crystallinity and crystal size have important effects on the properties of polymers. In previous studies, the increase of oxygen-containing functional groups results in a higher lattice defect density, which may lead to a decrease in surface crystallinity of polymer compounds According to the peak fit results of functional groups, oxygen-containing functional groups in PP-1 indicated an overall upward trend (1652 cm^−1^, 1737 cm^−1^ and 3480 cm^−1^) compared with PP-2 (1651 cm^−1^, 1741 cm^−1^ and 3511 cm^−1^). Therefore, PP-2 had a higher surface crystallinity. In addition, the content of aromatic hydrocarbons (benzene rings) in PP-2 (1539 cm^−1^) increased compared with PP-1 (1540 cm^−1^), indicating that the surface structure of PP-2 was highly ordered. The results demonstrated that the smaller the particle size, the higher-ordered the surface structure will be. IR could not be used as the main basis to judge the crystallinity of PP, so XRPD data of PP would be analyzed for supporting.

##### XRPD

XRPD was used to determine the crystal structure and properties of the molecule, in addition to the lattices in the molecule [40]. The XRPD diagrams of PP-1 and PP-2 are exhibited in Figure 9. Two major peaks, at 5.1° and 7.7°, were observed for PP-1 and PP-2, corresponding to its crystalline structure [41].

According to the log-scale of the original data, within the 2-Theta range of 14° to 26°, there was a “bulged” peak above the flat back and bottom, and it was judged that there was an amorphous characteristic peak. Therefore, this section of data was selected for peak fitting analysis. The XRPD diffraction spectrum and peak fitting diagram of PP-1 and PP-2 are displayed in Figure 10, and the area of each peak of PP-1 and PP-2 are given in Table 1. The peak fitting allowed us to distinguish the crystalline phase diffraction peaks (black solid line) from the amorphous characteristic peak (pink solid line). The area of each peak of PP-1 and PP-2 was portrayed in Table 2. The crystallinity formula (2) is as follows:(2)$$CrI=\frac{A_{c}}{A_{c}+A_{a}}\times 100\%$$
where *CrI* is the degree of crystallinity, *Ac* is the crystalline peak area, and *Aa* is the amorphous peak area. Results indicated that the crystallinity of PP-1 was 74.8%, and that of PP-2 was 76.9%. The results also indicated that PP-1 had a higher amorphous content than PP-2. Due to randomness in molecular conformation, amorphous systems were also called disordered systems [42]. Therefore, the higher-ordered surface structure of PP-2 resulted in higher crystallinity than PP-1.

### 3.2. Characterization of Test Formulations

#### 3.2.1. Analysis of Particle Size

Utilizing the particle size and distribution range of Invega Sustenna^®^ as a reference, the formulations were prepared by wet milling using either PP-1 or PP-2 samples. The particle size distribution image is exhibited in Figure 11, and the particle size data are displayed in Table 2. The distribution of particle size was evaluated by span value. The span values of formulations containing PP-1, formulations containing PP-2, and Invega Sustenna^®^ were 2.513, 2.597, and 2.457 μm, respectively. The smaller the span value is, the more uniform the particle size is. The particle size of formulations ranged from nanometer to micrometer. Such particle size distribution was consistent with the characteristics of sustained-release formulations.

#### 3.2.2. Evaluation of Dissolution

##### In Vitro Release

The dissolution behaviors of formulations containing PP-1 and formulations containing PP-2 and Invega Sustenna^®^ are displayed in Figure 12. Indeed, the formulation containing PP-1 and the formulation containing PP-2 and Invega Sustenna^®^ were approximately completely dissolved within 90 min, and the particle size distribution was consistent with dissolution behavior. The formulation containing PP-2 with a slightly larger particle size of d (0.9) had a smaller surface area than the formulation containing PP-1 and Invega Sustenna^®^ [43]. Therefore, the formulation containing PP-2 demonstrated a slightly slower late dissolution than the two other formulations.

##### Analysis of Similarity Factor (*f_2_*)

The characterization of dissolution profiles is compared by dissolution efficiency and the fit factor (*f_2_*) [44]. According to the guidelines of the FDA, the range of *f_2_* should be 50 to 100 [45]. The comparison of two test formulation dissolution profiles of Invega Sustenna^®^ is represented in Table 3. The similarity factors between the two formulations and Invega Sustenna^®^ were 85 and 72 respectively, and both values were greater than 50. It was indicated that the dissolution behaviors of two test formulations were similar to that of Invega Sustenna^®^.

### 3.3. Stability of Test Formulations

The particle size of nanomedicine is related to the degree of stability or changes i the quality of drug. Therefore, the particle sizes and distribution of nanomedicines have important influence on their quality and pharmacodynamic effects and are also important quality control indexes of nanomedicine. Therefore, the stability was assessed in terms of particle size. The physical stability of test formulations kept at room temperature within 30 days is displayed in Figure 13. According to the standard particle size range, the particle sizes of d (0.1), d (0.5), and d (0.9) were 0.13~0.30 μm, 0.60~1.00 μm, and 2.00~3.00 μm, respectively. The results indicated that the particle size of formulations containing PP-2 kept stability within 30 days; however, formulations containing PP-1 were always increasing, and the particle size of d (0.9) was out of standard on the 30th day. It was observed that the deposition of formulation containing PP-1 was more serious by observing its state at 30 days. Conversely, the suspension state of formulations containing PP-2 was excellent.

In micro-nano suspension, drugs are prone to sedimentation, aggregation, degradation, and denaturation, which leads to the instability. Micro-nano suspension is a thermodynamically unstable system with small particle size and large specific surface area, which produces high surface energy. Therefore, micro-nano particles tend to aggregate with each other to reduce the surface energy of the whole system [46]. The system can be unstable in the presence of a small number of amorphous nanoparticles. This is due to the fact that the molecular solubility of amorphous nanoparticles is usually at least one order of magnitude higher than the corresponding crystalline solubility. The amorphous particles dissolve spontaneously, while the crystalline particles grow, which is a combined process similar to Ostwald maturation [47]. According to the results of DSC, IR, and XRPD, the amorphous content of PP-1 was higher, which was responsible for the instability of formulation containing PP-1 compared with formulation containing PP-2. Conversely, the contact angle indicated that PP-2 had a higher SFE. This caused PP-2 to be more easily wetted, resulting in a uniform dispersion.

### 3.4. Pharmacokinetic In Vivo

A plasma concentration versus time profile of formulations containing PP-1 and formulations containing PP-2 and Invega Sustenna^®^ is presented in Figure 14. The pharmacokinetic parameters are available in Table 4. Following single-dose i.m. administration to rats, the release of paliperidone lasted for nearly 1 month. The T_max_ values of formulations containing PP-1 and formulations containing PP-2 and Invega Sustenna^®^ were 216 h. The C_max_ values of formulations containing PP-1 and formulations containing PP-2 and Invega Sustenna^®^ were 1428 ± 460, 1902 ± 125, and 1905 ± 296 ng/mL, respectively. The AUC_0–t_ of them were 642,602 ± 107,866, 498,846 ± 35,062, and 499,383 ± 89,611 ng/mL·h, respectively. Conversely, the AUC_0–∞_ of them were 918,380 ± 242,998, 526,208 ± 48,097, and 533,923 ± 81,653 ng/mL·h, respectively. Accordingly, the MRT_0–t_ of formulations of PP-2 and Invega Sustenna^®^ were similar to the MRT_0–∞_. Inversely, there was a great difference between the MRT_0–t_ and MRT_0–∞_ of formulations containing PP-1.

The results of pharmacokinetic parameters indicated that formulations containing PP-2 and Invega Sustenna^®^ were almost metabolized in rats within one month, but formulations containing PP-1 were not completely metabolized. In addition, there was a burst release phenomenon of formulations containing PP-1 in rats, which can be observed from plasma concentration versus time profile. In conclusion, the pharmacokinetic profiles and parameters of formulations containing PP-2 and Invega Sustenna^®^ tended to be consistent, but those of formulations containing PP-1 were significantly different from those of formulations containing PP-2 or Invega Sustenna^®^.

Undoubtedly, the preparation technology and dosage of auxiliary of the two test formulations were the same. The only difference was the crystallinity of PP-1 and PP-2. Previous study has indicated that an amorphous structure can enhance the absorption and performance of poorly soluble drugs and offer potential increases in solubility and biological activity of many thousands-fold compared with more crystalline forms of the drug [48]. Therefore, the amorphous part in formulations may have a rapid absorption and metabolism process, and the phenomenon is more obvious with higher amorphous content. DSC, IR, and XRPD results demonstrated together that PP-1 had a higher amorphous content than that of PP-2. Thus, formulations containing PP-1 were more prone to burst release phenomenon than formulations containing PP-2 in rats. Microscopic changes in the PP led to significant changes in the metabolism of the formulation in vivo, so the influence of crystallinity on drug safety should be considered in future studies.

## 4. Conclusions

In this paper, the nature of PP and the evaluation of formulations containing PP in vitro and in vivo were conducted to examine the significance of different crystallization processes. PPs, namely PP-1 and PP-2, were obtained from different crystallization processes. Contact angle results indicated that PP-2 had a higher SFE than PP-1. SEM results indicated that the difference in crystallization temperature can affect the appearance of PP. The saturation of PP-1 increased due to the higher cooling rate and more small particles were produced, so the particle size of PP-1 was smaller than that of PP-2. DSC, IR, and XRPD results suggested that the particle size of PP can influence the melting point, surface structure, and crystallinity, and the crystallinity of PP-2 was higher than that of PP-1. Taking the Invega Sustenna^®^ as reference, the formulation was prepared using either a PP-1 or PP-2 sample. Formulations containing PP-2 had a more stable and better suspension condition due to the higher crystallinity and SFE. In addition, the higher amorphous content of PP-1 caused the burst release phenomenon of formulations containing PP-1 in rats. Micro change determines macro change. Therefore, the results of this study also remind us to pay attention to the microscopic changes of the drug itself in the process of drug development.

## Figures and Tables

**Figure 1 pharmaceutics-14-01094-f001:**
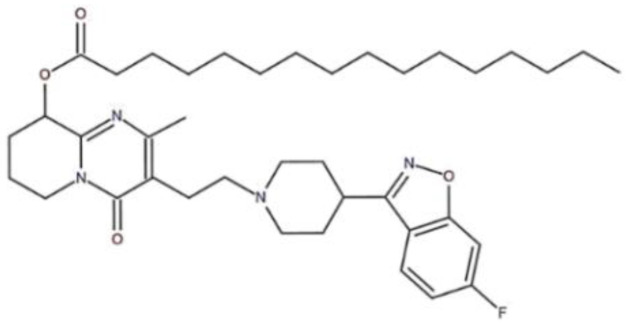
The chemical structure of paliperidone palmitate.

**Figure 2 pharmaceutics-14-01094-f002:**
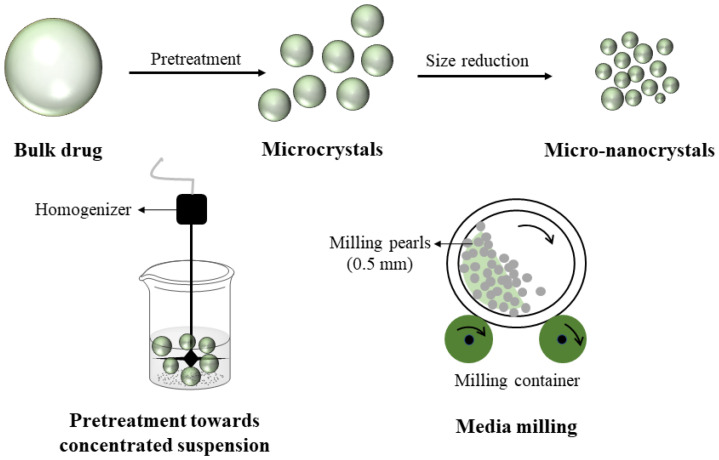
The preparation process of formulation containing paliperidone palmitate.

**Figure 3 pharmaceutics-14-01094-f003:**
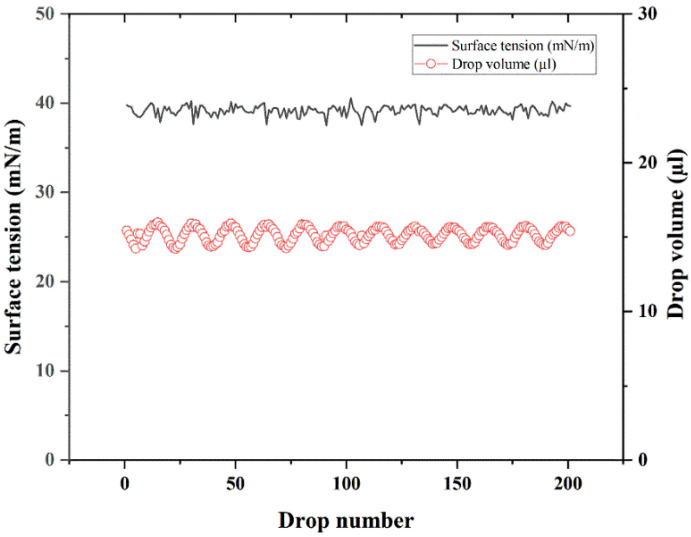
Surface tension of Tween 20 solution.

**Figure 4 pharmaceutics-14-01094-f004:**
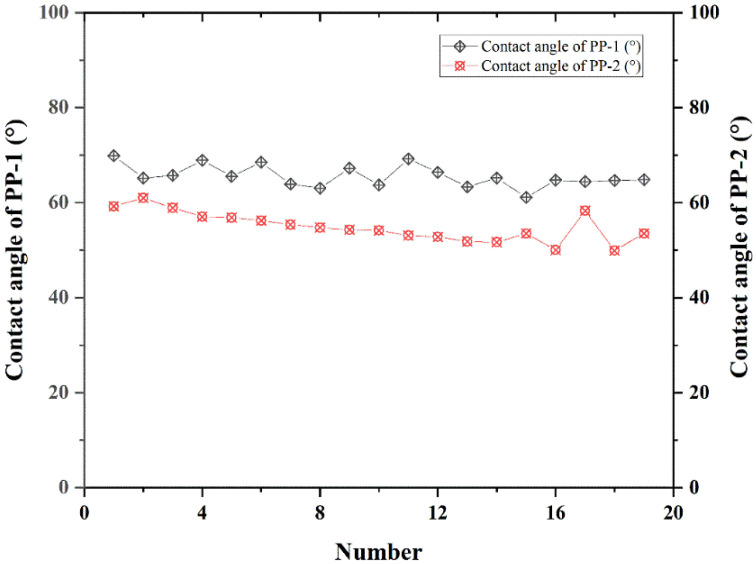
Contact angle of Tween 20 solution on PP-1 and PP-2.

**Figure 5 pharmaceutics-14-01094-f005:**
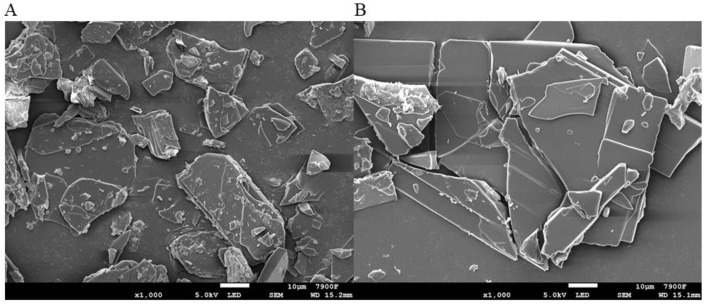
Morphologies of PP-1 (**A**) and PP-2 (**B**).

**Figure 6 pharmaceutics-14-01094-f006:**
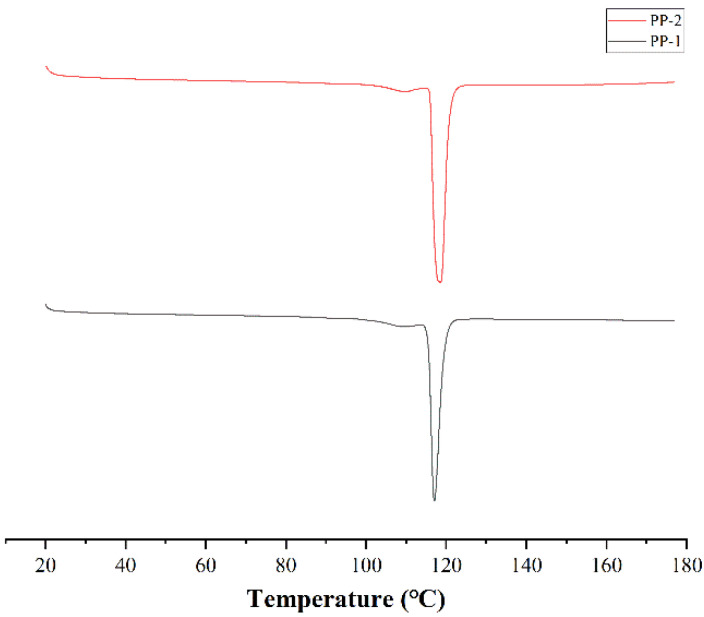
Thermograms of PP-1 and PP-2.

**Figure 7 pharmaceutics-14-01094-f007:**
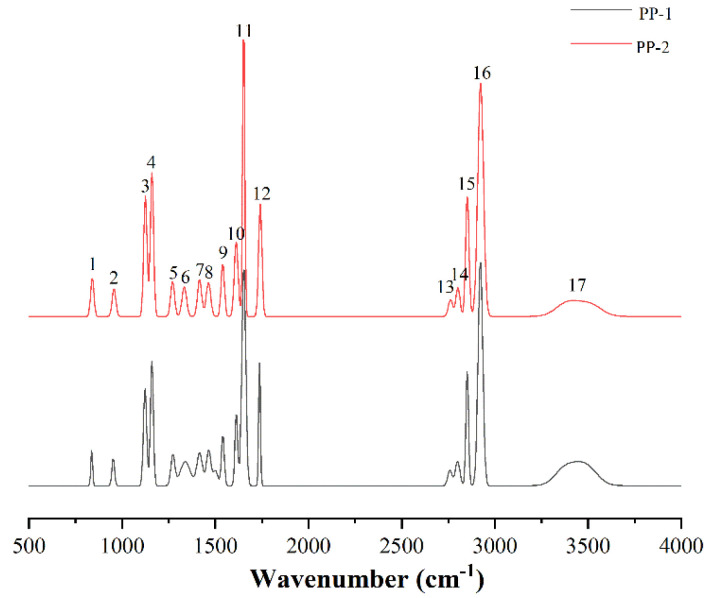
The IR diagrams of PP-1 and PP-2.

**Figure 8 pharmaceutics-14-01094-f008:**
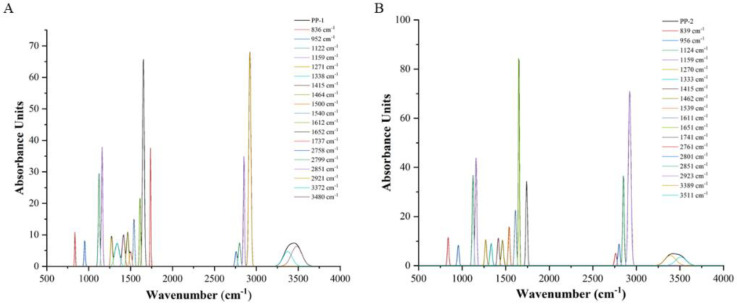
Fitting diagrams of each functional group in PP-1 (**A**) and PP-2 (**B**).

**Figure 9 pharmaceutics-14-01094-f009:**
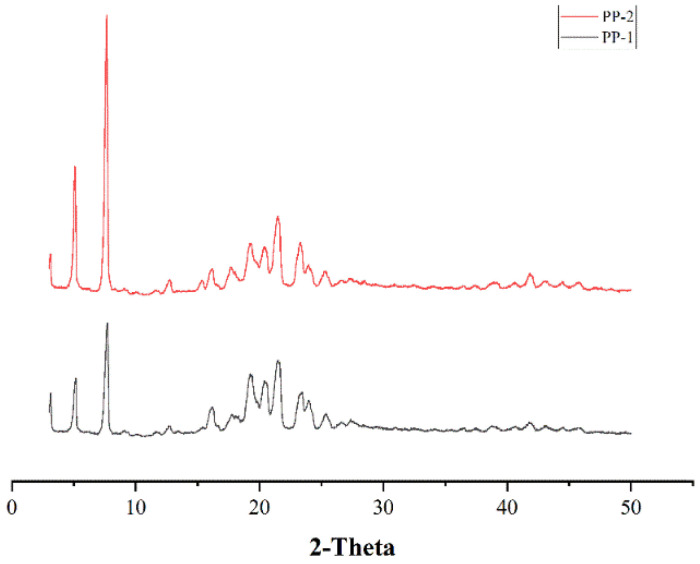
The XRPD diagrams of PP-1 and PP-2.

**Figure 10 pharmaceutics-14-01094-f010:**
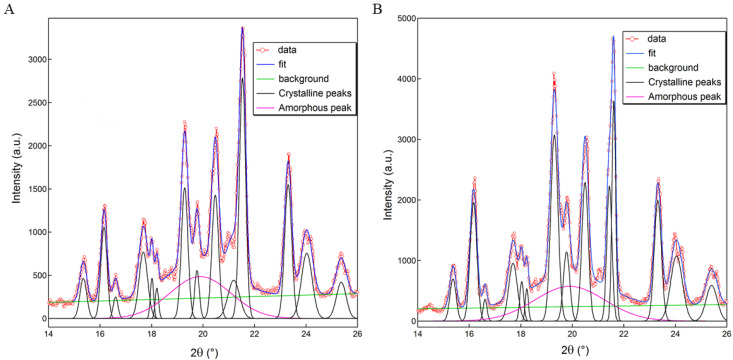
XRPD diffraction spectrum and peak fitting diagram of PP-1 (**A**) and PP-2 (**B**).

**Figure 11 pharmaceutics-14-01094-f011:**
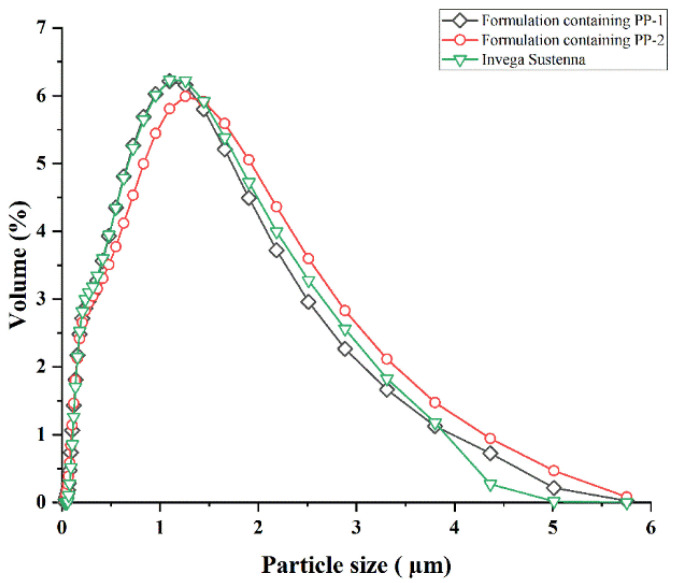
Particle size distribution image of formulations containing PP-1, formulations containing PP-2 or Invega Sustenna^®^.

**Figure 12 pharmaceutics-14-01094-f012:**
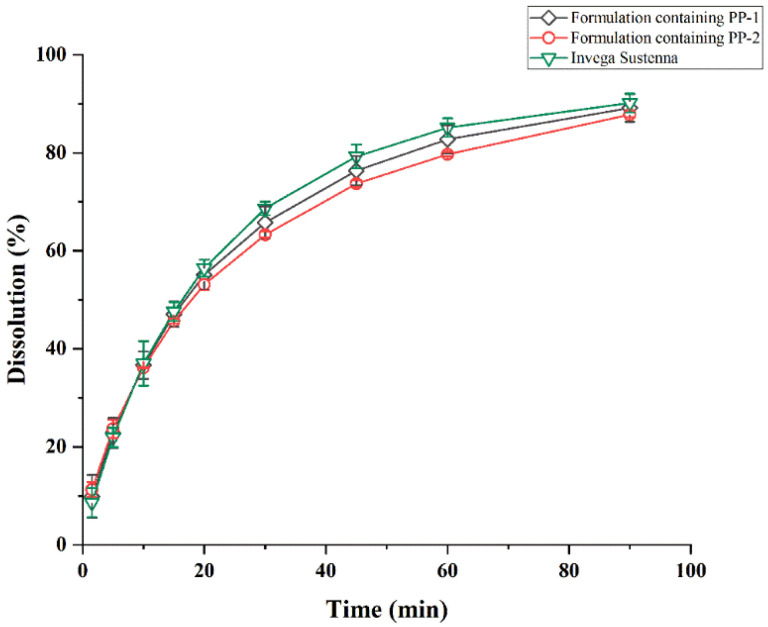
Dissolution profiles of formulations containing PP-1, formulations containing PP-2 and Invega Sustenna^®^ (n = 3).

**Figure 13 pharmaceutics-14-01094-f013:**
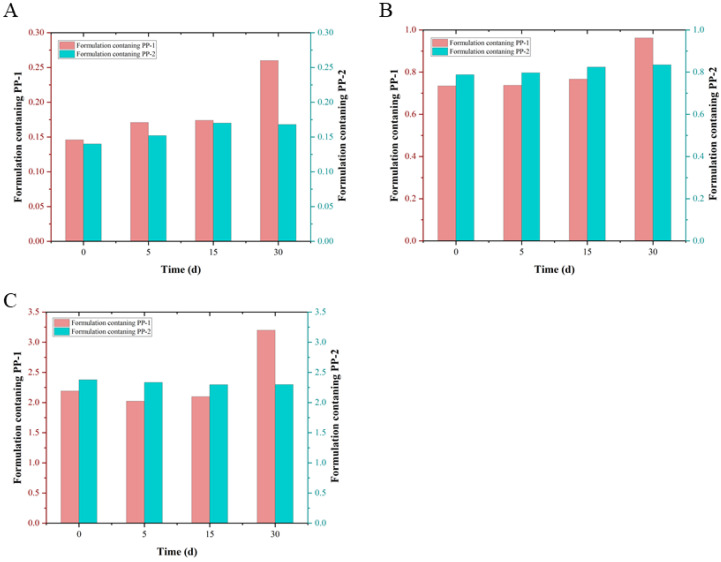
Physical stability of test formulations kept at room temperature within 30 days. (**A**) the particle sizes of d (0.1) within 30 days; (**B**) the particle sizes of d (0.5) within 30 days; (**C**) the particle size of d (0.9) within 30 days.

**Figure 14 pharmaceutics-14-01094-f014:**
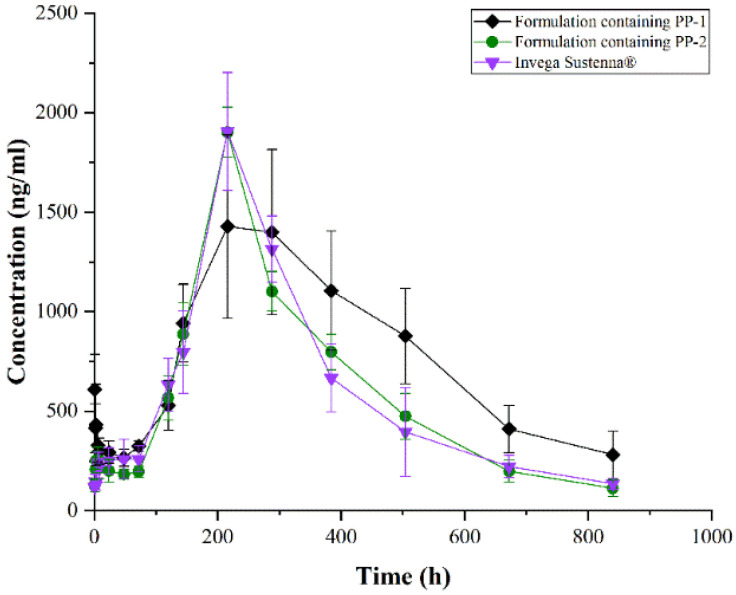
Plasma concentration versus time profile of formulations containing PP-1, formulations containing PP-2 or Invega Sustenna^®^ (n = 5).

**Table 1 pharmaceutics-14-01094-t001:** The area of each peak of PP-1 and PP-2.

Peak Number	Location of PP-1	Area	Location of PP-2	Area
1	15.35	154.13	15.37	211.21
2	16.15	316.33	16.16	644.36
3	16.60	56.93	16.61	67.75
4	17.68	317.83	17.70	397.21
5	18.01	71.79	18.04	125.46
6	18.21	57.33	18.23	80.90
7	19.90	1437.40	19.90	1959.10
8	19.29	477.00	19.30	1091.10
9	19.76	125.69	19.78	352.45
10	20.47	463.88	20.49	720.66
11	21.19	250.30	21.44	557.91
12	21.53	847.53	21.61	660.21
13	23.30	535.30	23.32	688.01
14	24.03	395.63	24.04	610.21
15	25.37	198.73	25.41	317.66

**Table 2 pharmaceutics-14-01094-t002:** Particle size of formulations containing PP-1, formulations containing PP-2 and Invega Sustenna^®^.

Sample Name	Span	d (0.1)/µm	d (0.5)/µm	d (0.9)/µm
Formulation containing PP-1	2.513	0.174	0.767	2.100
Formulation containing PP-2	2.597	0.168	0.825	2.310
Invega Sustenna^®^	2.457	0.188	0.785	2.118

**Table 3 pharmaceutics-14-01094-t003:** Comparison of two formulation release profiles of Invega Sustenna^®^.

Time (min)	Drug Dissolution (%)			
	Invega Sustenna^®^	Formulation Containing PP-1	Invega Sustenna^®^	Formulation Containing PP-2
1.5	8.63	9.87	8.63	11.30
5	21.92	22.83	21.92	23.67
10	37.03	36.66	37.03	36.15
15	47.62	47.06	47.62	45.68
20	56.46	55.09	56.46	53.08
30	68.66	65.80	68.66	63.30
45	79.27	76.37	79.27	73.72
60	85.14	82.73	85.14	79.72
90	90.17	89.14	90.17	87.81
*f* _2_	85		72	

**Table 4 pharmaceutics-14-01094-t004:** The lists of pharmacokinetic parameters of formulations containing PP-1, formulations containing PP-2 and Invega Sustenna^®^ (n = 5, mean ± SD).

Pharmacokinetic Parameter	Formulation Containing PP-1	Formulation Containing PP-2	Invega Sustenna^®^
C_max_ (ng/mL)	1428 ± 460	1902 ± 125	1905 ± 296
AUC_0–t_ (ng/mL·h)	642,602 ± 107,866	498,846 ± 35,062	499,383 ± 89,611
AUC_0–∞_ (ng/mL·h)	918,380 ± 242,998	526,208 ± 48,097	533,923 ± 81,653
MRT_0–t_ (h)	376 ± 23	325 ± 14	321 ± 7
MRT_0–∞_ (h)	822 ± 475	363 ± 30	378 ± 46

## Data Availability

The data presented in this study are available on request from the corresponding author.

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
