# Peer review of "Comparison of Paliperidone Palmitate from Different Crystallization Processes and Effect on Formulations In Vitro and In Vivo"

_pharmaceutics, 2022, doi:10.3390/pharmaceutics14051094_

Round 1

Reviewer 1 Report

The paper is focused on the detailed morpho-structural analysis of paliperidone palmitate (PP), a second generation antipsychotic used in the drug treatment of schizophrenia. The article is coherent and the obtained data  are most cases correlated. However, there are some information that needs to be explained.

1.The first sentence of the Introduction should be reworded. The term, "the treatment of schizophrenia is repeated" twice.

2.The crystallization process is not visible in the DSC. Could you explain that?

3.The conclusions chapter needs to be improved. In this form it does not reflect all the  obtained results.

Author Response

Dear Reviewer:

Thank you for your comments concerning our manuscript entitled “Comparison of paliperidone palmitate from different crystallization processes and effect on formulations in vitro and in vivo” (Manuscript ID: pharmaceutics-1708708). Those comments are all valuable and helpful for revising and improving our paper, as well as the important guiding significant to our researches. We have studied comments carefully and have made correction which we hope meet the approval. The red font indicates the detailed modifications in manuscript. The main corrections in the paper and responds to the reviewers′ comments are as flowing:

Point 1: The first sentence of the Introduction should be reworded. The term, "the treatment of schizophrenia is repeated" twice.

Response 1: Thank you for your suggestion. The sentence had been reworded in manuscript. (Line 36-37)

Point 2: The crystallization process is not visible in the DSC. Could you explain that?

Response 2: Thank you for your question. For the same drug, the higher the melting point, the higher the crystallinity. And the conclusion had been confirmed via IR and XRPD. The IR result showed that PP-2 had a more order surface structure. The more order surface structure, the higher crystallinity. The XRPD semiquantitative results also suggested that PP-2 had a higher crystallinity. According your comment, we made some changes in red in manuscript. (Section 3.1.3.1)

Point 3: The conclusions chapter needs to be improved. In this form it does not reflect all the obtained results.

Response 3: Considering your suggestion, we improved the conclusions chapter. The modifications have been made in red in manuscript. (Section 4)

Thank you for your detailed reading of my manuscript. We tried our best to improve the manuscript and the changes have marked in red in manuscript.

Once again, thank you very much for your comments and suggestions. Looking forward to hearing from you.

Sincerely,

Junfeng Shi

Reviewer 2 Report

The authors should more comment about the reasons why little variations in cristallinity can affect the stability and quite relevant differences in the metabolic behavior of formulations in vivo.

Is the cristallization procedure reproducible in terms of cristallinity? The auhtors should also more comment about cristallinity of PP1 and PP2

Author Response

Dear Reviewer:

Thank you for your comments concerning our manuscript entitled “Comparison of paliperidone palmitate from different crystallization processes and effect on formulations in vitro and in vivo” (Manuscript ID: pharmaceutics-1708708). Those comments are all valuable and helpful for revising and improving our paper, as well as the important guiding significant to our researches. We have studied comments carefully and have made correction which we hope meet the approval. The red font indicates the detailed modifications in manuscript. The main corrections in the paper and responds to the reviewers′ comments are as flowing:

Point 1: The authors should more comment about the reasons why little variations in crystallinity can affect the stability and quite relevant differences in the metabolic behavior of formulations in vivo.

Response 1: Considering your suggestion, we made some changes in red in manuscript. (Section 3.3 and 3.4)

Point 2: “Is the crystallization procedure reproducible in terms of crystallinity? The authors should also more comment about crystallinity of PP-1 and PP-2.”

Response 2: Thank you for your question and suggestion. The crystallization procedures have been reworded in red in manuscript. (Section 2.2)

Thank you for your detailed reading of my manuscript. We tried our best to improve the manuscript and the changes have marked in red in manuscript.

Once again, thank you very much for your comments and suggestions. Looking forward to hearing from you.

Sincerely,

Junfeng Shi

Reviewer 3 Report

The manuscript by Shi and coworkers intitled “Comparison of paliperidone palmitate from different crystallization processes and effect on formulations in vitro and in vivo” deals with physico-chemical and biological evaluation of new paliperidone palmitate formulations prepared by milling and compared to the related marketed product, namely Invega Sustenna. The experiments are exhaustive, and the results obtained are clearly presented. However the different results obtained should be deeply interpreted, especially regarding results that converge to the same interpretation. Furthermore some references dealing with the state of the art of pharmaceutical solid forms should be introduced in the manuscript. Therefore, the authors should take into consideration the suggestions proposed hereafter for considering publication in Pharmaceutics.

Page 1, Line 17: “PP-1 and PP-2 were prepared” should be “Two different formulations of PP, namely PP-1 and PP-2 were prepared”.

Page 1, Lines 19 – 20: The word “respectively” is not convenient. I would rather say “using either PP-1 or PP-2 sample.”.

Page 1, Line 28: “PP-2 and Invega” should be “PP-2 or Invega”.

Page 1, Lines 37 – 38: The authors introduce, without mentioning it, prodrug property. Therefore I would rather say: “PP, a prodrug of paliperidone, hydrolyzes…”.

Page 2, Line 59: “is an injectable aqueous suspension formulation” should be “are injectable aqueous suspension formulations”.

Page 2, Line 86: “In present study” should be “In the present study”.

Page 4, Section 2.3.3.3: Since some peak area of the X-rays patterns will be analyzed in the Results and discussion part, the authors should indicate here the acquisition time of each spectrum and, if it is the case, specify that the spectra were normalized as function of the acquisition time.

Page 4, Line 156: “was demonstrated” should be “is proposed”. Please replace in the whole manuscript the past tense by the present one when mentioning a figure or a table that is actually in the present manuscript.

Page 4, Line 158: Here again, the word “respectively” is not convenient. I would rather say “were prepared as described in Section 2.2.”.

 Page 4, Figure2: It suggest the authors to indicate the nature of the bulk drug phase. Is that solubilized drug? In which solvent? Or drug in the powder form?

Page 6, Line 246: “39.76” should be replaced by “39.8”.

Page 6, Line 247: “reduce” should be “decrease”.

Page 7, Line 288: “117.17 °C and 118.61 °C” should be “117.2 °C and 118.6 °C”.

Pages 7 – 8, Lines 290 – 291: The authors claim that “The date (DSC data) indicated that PP-1 and PP-2 had no crystal transformation”. What do they want to say exactly? Furthermore, the other should develop the DSC part by explaining the small depletion of melting point for PP-1 compared to PP-2. There is some data in the literature showing, especially for organic drug, that the depletion in melting is link to size reduction of the powder.

Page 9, Line 332: The higher ordered surface structure of PP-2 can be link here to the size reduction of PP-1 demonstrated via the DSC results (see above).

Page 9, Lines 337 – 338: This sentence should be rewritten.

Pages 9 – 10, Lines 346 – 348: Please replace the sentence by the following: “The peak fitting allowed us to distinguish the crystalline phase diffraction peaks (black solid line) from the amorphous characteristic peak (pink solid line)”.

Page 10, Lines 351 – 352: The authors should develop this part dealing with crystallinity by comparing their findings with the previous ones (that obtained from morphology, DSC, and/or IR results).

Page 11, Line 361: Here again, the word “respectively” is not convenient. I would rather say “using either PP-1 or PP-2 sample.”.

Page 11, Line 380: “The formulation containing” should be “Indeed, the formulation containing”.

Page 11, Lines 382 – 383: The sentence should be “Therefore, the formulation containing PP-2 showed a slight slower late dissolution than the two other formulations.”.

Page 14, Figure 14: The error bars cannot be distinguished. I suggest to increase the thickness of the lines.

Page 14, Line 448: “PP2 and Invega” should be “PP2 or Invega” .

Page 14, Figure 14: The caption should be: “Plasma concentration versus time profile of formulation containing PP-1, PP-2 or Invega…”. Please complete.

Page 14, Table 5: Do the authors are sure about their MRT0-inf values? I The discrepancy between that of PP-1 and those for PP-2 and Invega Sustenna does not seem to fit to the data showed on Figure 14.

Author Response

Dear Reviewer:

Thank you for your letter and for the reviewers′ comments concerning our manuscript entitled “Comparison of paliperidone palmitate from different crystallization processes and effect on formulations in vitro and in vivo” (Manuscript ID: pharmaceutics-1708708). Those comments are all valuable and helpful for revising and improving our paper, as well as the important guiding significant to our researches. We have studied comments carefully and have made correction which we hope meet the approval. The red font indicates the detailed modifications in manuscript. The main corrections in the paper and responds to the reviewers′ comments are as flowing:

Point 1: Page 1, Line 17: “PP-1 and PP-2 were prepared” should be “Two different formulations of PP, namely PP-1 and PP-2 were prepared”.

Response 1: Thank you for your comment. The detail has been revised in manuscript.

Point 2: Page 1, Lines 19-20: The word “respectively” is not convenient. I would rather say “using either PP-1 or PP-2 sample.”.

Response 2: The detail has been revised in manuscript.

Point 3: Page 1, Line 28: “PP-2 and Invega” should be “PP-2 or Invega”.

Response 3: The detail has been revised in manuscript.

Point 4: Page 1, Lines 37-38: The authors introduce, without mentioning it, prodrug property. Therefore, I would rather say: “PP, a prodrug of paliperidone, hydrolyzes…”.

Response 4: The detail has been revised in manuscript.

Point 5: Page 2, Line 59: “is an injectable aqueous suspension formulation” should be “are injectable aqueous suspension formulations”.

Response 5: The detail has been revised in manuscript.

Point 6: Page 2, Line 86: “In present study” should be “In the present study”.

Response 6: The detail has been revised in manuscript.

Point 7: Page 4, Section 2.3.3.3: Since some peak area of the X-rays patterns will be analyzed in the Results and discussion part, the authors should indicate here the acquisition time of each spectrum and, if it is the case, specify that the spectra were normalized as function of the acquisition time.

Response 7: Considering your suggestion, we made some changes in red in manuscript. (Section 2.3.3.3)

Point 8: Page 4, Line 156: “was demonstrated” should be “is proposed”. Please replace in the whole manuscript the past tense by the present one when mentioning a figure or a table that is actually in the present manuscript.

Response 8: Details have been revised in manuscript.

Point 9: Page 4, Line 158: Here again, the word “respectively” is not convenient. I would rather say “were prepared as described in Section 2.2.”.

Response 9: The detail has been revised in manuscript.

Point 10: Page 4, Figure2: It suggest the authors to indicate the nature of the bulk drug phase. Is that solubilized drug? In which solvent? Or drug in the powder form?

Response 10: Thank you for your question. PP was characterized as practically insoluble in water (intrinsic solubility below 0.1 µg/ml), so it was prepared into suspension by wetting agent. (Section 2.4)

Point 11: Page 6, Line 246: “39.76” should be replaced by “39.8”.

Response 11: The detail has been revised in manuscript.

Point 12: Page 6, Line 247: “reduce” should be “decrease”.

Response 12: The detail has been revised in manuscript.

Point 13: Page 7, Line 288: “117.17 °C and 118.61 °C” should be “117.2 °C and 118.6 °C”.

Response 13: Details have been revised in manuscript.

Point 14: Pages 7-8, Lines 290-291: The authors claim that “The date (DSC data) indicated that PP-1 and PP-2 had no crystal transformation”. What do they want to say exactly? Furthermore, the other should develop the DSC part by explaining the small depletion of melting point for PP-1 compared to PP-2. There is some data in the literature showing, especially for organic drug, that the depletion in melting is link to size reduction of the powder.

Response 14: Considering your suggestions, we recognized our own shortcomings. Therefore, we added some content in Section 3.1.3.1.

Point 15: Page 9, Line 332: The higher ordered surface structure of PP-2 can be link here to the size reduction of PP-1 demonstrated via the DSC results (see above).

Response 15: Thank you for bringing such a good proposal. The modifications have displayed in manuscript. (Section 3.1.3.2)

Point 16: Page 9, Lines 337-338: This sentence should be rewritten.

Response 16: The sentence had been reworded in red in manuscript. (Section 3.1.3.2)

Point 17: Pages 9-10, Lines 346-348: Please replace the sentence by the following: “The peak fitting allowed us to distinguish the crystalline phase diffraction peaks (black solid line) from the amorphous characteristic peak (pink solid line)”.

Response 17: The detail has been revised in manuscript.

Point 18: Page 10, Lines 351-352: The authors should develop this part dealing with crystallinity by comparing their findings with the previous ones (that obtained from morphology, DSC, and/or IR results).

Response 18:

Point 19: Page 11, Line 361: Here again, the word “respectively” is not convenient. I would rather say “using either PP-1 or PP-2 sample.”.

Response 19: The detail has been revised in manuscript.

Point 20: Page 11, Line 380: “The formulation containing” should be “Indeed, the formulation containing”.

Response 20: The detail has been revised in manuscript.

Point 21: Page 11, Lines 382 - 383: The sentence should be “Therefore, the formulation containing PP-2 showed a slight slower late dissolution than the two other formulations.”.

Response 21: The detail has been revised in manuscript.

Point 22: Page 14, Figure 14: The error bars cannot be distinguished. I suggest to increase the thickness of the lines.

Response 22: Considering your suggestions, we replotted the image.

Point 23: Page 14, Line 448: “PP2 and Invega” should be “PP-2 or Invega”.

Response 23: The detail has been revised in manuscript.

Point 24: Page 14, Figure 14: The caption should be: “Plasma concentration versus time profile of formulation containing PP-1, PP-2 or Invega…”. Please complete.

Response 24: The detail has been revised in manuscript.

Point 25: Page 14, Table 5: Do the authors are sure about their MRT0-inf values? I The discrepancy between that of PP-1 and those for PP-2 and Invega Sustenna does not seem to fit to the data showed on Figure 14.

Response 25: Considering your question, we recalculated the pharmacokinetic parameters of each group and redrawn the image. (Section 3.4)

Thank you for your detailed reading of my manuscript. We tried our best to improve the manuscript and the changes have marked in red in manuscript.

Once again, thank you very much for your comments and suggestions. Looking forward to hearing from you.

Sincerely,

Junfeng Shi

Reviewer 4 Report

Authors report on the effect of different crystallization processes for preparation of paliperidone palmitate and in vitro and in vivo bioassay. While the aim of the work is clear and well performed, the main concern is regarding the English – much improvement is needed.

Page 1, Lines 35-37: sentence should be rewritten, the same info is given twice in the same sentence.

Page 2, Line 72: what is meant by “a synthetic monocrystal drug”? For sure it shouldn’t be one crystal…

Page 3, Section 2.2 should be rewritten. “the last of PP synthesis was solvent crystallization to obtain pure drug” – crystallization is not a synthesis. Better it would be stated: “the last step of PP preparation was solvent crystallization to obtain pure drug”. It is not clear if the sentence “First, the crude product of PP was dissolved and crystallized in ethyl acetate.” is an explanation of a previous sentence or is an explanation of further processes.

Page 3, Section 2.2: Sentences “Next, the filter cake was dissolved again and crystallized in ethanol.“ and “Finally, the product was cooled, filtered and dried.“  give unclear information. Is the last sentence implying that another crystallization was performed? If both sentences are describing one crystallization than they should be merged into one sentence.

Page 3, Section 2.2: Why the order of explanation is first PP-2 and then PP-1? Reverse the order. The word “nevertheless” is redundant and could cause confusion.

Figure 2, bottom-right picture: shouldn’t both dark green circles rotate in the same direction if the main container is supposed to rotate too?

Section 3: when explaining figures, a present tense should be used (not “was exhibited” but “is presented”), also when discussing on IR and other data (not “was related to” but “is related to”; etc.)

Author Response

Dear Reviewer:

Thank you for your letter and for the reviewers′ comments concerning our manuscript entitled “Comparison of paliperidone palmitate from different crystallization processes and effect on formulations in vitro and in vivo” (Manuscript ID: pharmaceutics-1708708). Those comments are all valuable and helpful for revising and improving our paper, as well as the important guiding significant to our researches. We have studied comments carefully and have made correction which we hope meet the approval. The red font indicates the detailed modifications in manuscript. The main corrections in the paper and responds to the reviewers′ comments are as flowing:

Point 1: Page 1, Lines 35-37: sentence should be rewritten, the same info is given twice in the same sentence.

Response 1: Considering your suggestions, we recognized our own shortcomings. We have revised the sentence in red in manuscript. (Section 1.)

Point 2: Page 2, Line 72: what is meant by “a synthetic monocrystal drug”? For sure it shouldn’t be one crystal…

Response 2: Considering your question, we realized the error of this statement. What we want to express is the crystallization process of PP rather than the synthesis process. So, we have modified the sentence in red in manuscript. (Line 71)

Point 3: Page 3, Section 2.2 should be rewritten. “the last of PP synthesis was solvent crystallization to obtain pure drug” — crystallization is not a synthesis. Better it would be stated: “the last step of PP preparation was solvent crystallization to obtain pure drug”. It is not clear if the sentence “First, the crude product of PP was dissolved and crystallized in ethyl acetate.” is an explanation of a previous sentence or is an explanation of further processes.

Response 3: I am glad for you brought up the suggestion. Section 2.2 has been amended in accordance with your suggestions.

Point 4: Page 3, Section 2.2: Sentences “Next, the filter cake was dissolved again and crystallized in ethanol.” and “Finally, the product was cooled, filtered and dried.” give unclear information. Is the last sentence implying that another crystallization was performed? If both sentences are describing one crystallization than they should be merged into one sentence.

Response 4: Considering your suggestion, we recognized our own shortcoming. Therefore, we reworded the Section 2.2 in red in manuscript.

Point 5: Page 3, Section 2.2: Why the order of explanation is first PP-2 and then PP-1? Reverse the order. The word “nevertheless” is redundant and could cause confusion.

Response 5: Thank you for your comment. We have modified in Section 2.2.

Point 6: Figure 2, bottom-right picture: shouldn’t both dark green circles rotate in the same direction if the main container is supposed to rotate too?

Response 6: Thank you for your careful reading. We have modified Figure 2.

Point 7: Section 3: when explaining figures, a present tense should be used (not “was exhibited” but “is presented”), also when discussing on IR and other data (not “was related to” but “is related to”; etc.)

Response 7: Thank you for your comment. The details have been revised in red in manuscript.

Thank you for your detailed reading of my manuscript. We tried our best to improve the manuscript and the changes have marked in red in manuscript.

Once again, thank you very much for your comments and suggestions. Looking forward to hearing from you.

Sincerely,

Junfeng Shi

Round 2

Reviewer 3 Report

The authors have taken into consideration most of the comments made. Nevertheless, some minor improvements of their article are needed prior publication in Pharmaceutics.

In the whole manuscript: some references dealing with the state of the art of pharmaceutical solid forms should be introduced in order to center the objective of such a work.

Page 3, Lines 111 – 122: “all solids” should be “all solid particles” everywhere in this paragraph. Moreover, the paragraph should be rewritten since it is not understandable at this stage. Finally, the pore size for each filtration performed should be specified.

Page 8, Lines 295 – 298: The reference 24 proposed by the authors has not been associated to a doi. The related article is hard to find but, I do not think it is suitable for organic compounds such as Paliperidone since the related journal is “Mining and Metallurgical Engineering”. Therefore, I suggest to the authors to replace “melting point of materials decreases” by “melting point of organic materials decreases” and replace reference 24 by suitable references dealing with depletion of melting point of organic active ingredients.

Page 8, Line 299: Replace “the further” by “further”.

Page 13, Line 446: “1427.55 ± 460.18, 1901.73 ± 125.45 and 1905.22 ± 296.22” should be “1 428 ± 460, 1 902 ± 125 and 1 905 ± 296”.

Page 14, Lines 446 – 447: “642602.20 ± 107866.30, 498845.75 ± 35061.58 and 499382.86 ± 89611.19” should be “642 602 ± 107 866, 498 846 ± 35 062 and 499 383 ± 89 611”.

Page 14, Lines 448 – 449: “918379.95 ± 242997.89, 526207.68 ± 48097.41 and 533923.20 ± 81652.92” should be “918 380 ± 242 998, 526 208 ± 48 097 and 533 923 ± 81 653”.

Page 14, Line 462: “Study” should be “Previous study”.

Page 14, Table 5: Please modify the data as proposed previously. All the data of this table should be modified the same way.

Page 15, Line 485: “than PP-2” should be “than that of PP-2”

Reference section: The authors should pay attention and standardize the reference with doi and correct abbreviation for all journals mentioned.

Author Response

Dear Reviewer:

Thank you for your leeter concerning our manuscript entitled “Comparison of paliperidone palmitate from different crystallization processes and effect on formulations in vitro and in vivo” (Manuscript ID: pharmaceutics-1708708). Those comments are all valuable and helpful for revising and improving our paper, as well as the important guiding significant to our researches. We have checked the manuscript and revised it according to the comments. The main corrections in the paper and responds to the reviewers′ comments are as flowing:

Point 1: In the whole manuscript: some references dealing with the state of the art of pharmaceutical solid forms should be introduced in order to center the objective of such a work.

Response 1: Considering your suggestion, we have enriched the part of Introduction. (Line 80-92)

Point 2: Page 3, Lines 111 - 122: “all solids” should be “all solid particles” everywhere in this paragraph. Moreover, the paragraph should be rewritten since it is not understandable at this stage. Finally, the pore size for each filtration performed should be specified.  

Response 2: Thank you for your question and suggestion. We reworded the section 2.2 in red in manuscript.

Point 3: Pag 8, Lines 295 - 298: The reference 24 proposed by the authors has not been associated to a doi. The related article is hard to find but, I do not think it is suitable for organic compounds such as Paliperidone since the related journal is “Mining and Metallurgical Engineering”. Therefore, I suggest to the authors to replace “melting point of materials decreases” by “melting point of organic materials decreases” and replace reference 24 by suitable references dealing with depletion of melting point of organic active ingredients.

Response 3: Considering your comment, we have corrected the relevant sentence and consulted the relevant literature to replace the reference 24.

Point 4: Page 8, Line 299: Replace “the further” by “further”.

Response 4: The detail has been revised.

Point 5: Page 13, Line 446: “1427.55 ± 460.18, 1901.73 ± 125.45 and 1905.22 ± 296.22” should be “1 428 ± 460, 1 902 ± 125 and 1 905 ± 296”.

Response 5: The detail has been revised.

Point 6: Page 14, Lines 446 – 447: “642602.20 ± 107866.30, 498845.75 ± 35061.58 and 499382.86 ± 89611.19” should be “642 602 ± 107 866, 498 846 ± 35 062 and 499 383 ± 89 611”.

Response 6: The detail has been revised.

Point 7: Page 14, Lines 448 – 449: “918379.95 ± 242997.89, 526207.68 ± 48097.41 and 533923.20 ± 81652.92” should be “918 380 ± 242 998, 526 208 ± 48 097 and 533 923 ± 81 653”.

Response 7: The detail has been revised.

Point 8: Page 14, Line 462: “Study” should be “Previous study”.

Response 8: The detail has been revised.

Point 9: Page 14, Table 5: Please modify the data as proposed previously. All the data of this table should be modified the same way.

Response 9: Thank you for your suggestion. The detail has been revised.

Point 10: Page 15, Line 485: “than PP-2” should be “than that of PP-2”.

Response 10: The detail has been revised.

Point 11: Reference section: The authors should pay attention and standardize the reference with doi and correct abbreviation for all journals mentioned.

Response 11: Considering your suggestion, we have examined all references.

Thank you for your reading of my manuscript again. We tried our best to improve the manuscript and the changes have marked in red in manuscript.

Thank you again for your valuable comments and suggestions. Looking forward to hearing from you.

Sincerely,

Junfeng Shi

This manuscript is a resubmission of an earlier submission. The following is a list of the peer review reports and author responses from that submission.

Round 1

Reviewer 1 Report

The paper proposes a complex study of the properties of paliperidone palmitate, an antipsychotic agent widely used in the treatment of schizophrenia. Although the work shows a considerable number of characterization methods, and the results are diverse, their correlation is missing. Each section is presented in a superficial way and key information are missing. For example:

  1. The results of the particle size analysis are missing.
  2.  X-ray diffraction is unclear, indexation of the peaks in relation to a database is missing.
  3. The interpretation of the DSC thermograms is erroneous, only the melting points of the two samples  are presented. What can be said about the crystallization processes? What is the connection between the shape of the samples and the variation of their melting point?

  4. The conclusions are modest and do not reflect the information presented.

Reviewer 2 Report

The manuscript deals with the preparation and characterization of two forms of Paliperidone obtained from two different cristallization processes. The manuscript is rich of data and materials sufficiently characterised both in vitro and in vivo.

The authors should better focus the aim of the manuscript in the introduction and the importance of investigating this topic.

In 2.2. Preparation of Paliperidone palmitate it is  not clear the process of crystallization. What is the crude product? How crystallization process was performed? Are there PP-1 and PP-2 two polimorph of the drug?

At which temperature surface tension measurements and wettability were performed?

Since a discussion section is lacking in the manuscript, conclusions should be implemented, highlighting the relevance of the findings.